# Risk Aware Negative Sampling in Link Prediction

## Abstract

It is commonly believed that Message Passing Neural Networks (MPNNs) struggle in link prediction settings due to limitations in their expressive power. Recent work has focused on developing more expressive model classes, which are capable of learning link representations through techniques such as labeling tricks, the inclusion of structural features, or the use of subgraph methods. These approaches have yielded significant performance improvements across a range of benchmark datasets. However, an interesting question remains: have we fully wrung out the performance by optimizing the other aspects of the training process? In this work, we present results that indicate that significant amounts of model performance have been left on the table by the use of easy negative-samples during training. We theoretically explore the generalization gap and excess risk to quantify the performance loss caused by easy negatives. Motivated by this analysis, we introduce Risk Aware Negative Sampling (RANS), which efficiently performs dynamic hard-negative-mining. Empirical results show that a simple GCN augmented by RANS realizes between 20% and 50% improvements in predictive accuracy when compared with the same model trained with standard negative samples.

## 1 Introduction

Link prediction is an important machine learning task that aims to predict unobserved edges connecting two vertices. Link prediction traditionally operates on graph-structured data, which is ubiquitous in industrial settings as it provides a natural way to represent entities and complex relationships between them (Chamberlain et al., 2023). For example, predicting new friendship relationships or post engagements can be framed as link prediction tasks (El-Kishky et al., 2022; Cai et al., 2023) in social media companies. This is accomplished by constructing an unsupervised set of positive samples, such as existing links, and learning a representation that reliably predicts those relationships to exist with higher probability than a set of negatives (Kumar et al., 2020; Yang et al., 2024).

Given its importance, link prediction has received broad interest both industrially and academically, with significant work dedicated to improving modeling modalities. The simplest approaches are heuristic methods, which provide structural measures for link similarity (Zhou et al., 2009; Adamic & Adar, 2003). Another class of popular methods aims to compute unsupervised node embeddings that minimize graph reconstruction error (Bordes et al., 2013; Kazemi & Poole, 2018; Rossi et al., 2021; Lerer et al., 2019). In recent years, this focus has shifted to modeling graphs using Message Passing Neural Networks (MPNNs)(Kipf & Welling, 2016; Hamilton et al., 2017). However, unlike node or graph-level tasks, MPNNs often struggle with link-level tasks. This is commonly attributed to two reasons: (1) MPNNs are unable to count triangles(Chen et al., 2020) because they are equivalent to the 1-WL test (Xu et al., 2018), and (2) MPNNs learn structural node representations when structural edge representations are required for link prediction (Srinivasan & Ribeiro, 2019).

These insights have inspired the development of more expressive MPNNs through the use of labeling tricks (Zhang et al., 2021), conversion from link prediction to subgraph classification (Zhang & Chen, 2018a; Wang & Zhang, 2021; Yin et al., 2023), or the inclusion of graph structural features (Chamberlain et al., 2023; Yun et al., 2021). While these advances have led to complicated models and impressive performance gains, an open question remains: are we extracting maximum performance from a given model, even if it's not maximally expressive? This question is particularly

important because more complicated models are often difficult to scale in industrial settings (Ma et al., 2022; Zheng et al., 2022; Borisyuk et al., 2024).

Due to the unsupervised nature of link prediction, there are only a few core components to the problem: the model, additional feature engineering (through labeling tricks or structural features), and the selection of negative samples. While both feature engineering and modeling techniques have received significant research attention, negative sampling has been comparatively less explored in the context of link prediction. The quality of negatives, however, is intuitively quite important. Consider the case of a friendship graph on a site like Facebook, where the task is to recommend potential friends whom users might know. A model will almost certainly be better informed about user relationships when trained with negatives that are close to their social location (e.g., the town where they live) rather than with uniform random sampling, which would likely generate trivial negative examples.

Previous graph-related work has found that the selection of hard negatives through either feature similarity (Pancha et al., 2022) or graph-structure sampling (Yang et al., 2020) leads to performance improvements in some graph learning settings. Beyond graphs and link prediction, hard negative mining has received significant attention from the computer vision (Xuan et al., 2020; Jin et al., 2018; Sun et al., 2019) and NLP communities (Zhang & Stratos, 2021; Dasgupta & Ng, 2009).

Based on our understanding of the importance of hard negatives, as well as the attention from other communities, we address the question: *"Can we learn better link prediction models by improving negative sampling methods?"* We answer this question first through the analysis of link prediction using the empirical risk minimization framework, which allows us to develop clear bounds for both the generalization gap and the excess risk in terms of the unsampled negatives. These theoretical insights inspire a simple negative resampling method that we term **Risk Aware Negative Sampling (RANS)**, which is highly flexible and applicable to any MPNN. To validate this method and our analysis, we evaluate the performance of RANS on a range of datasets and present our results in Section 7.

## 2 RELATED WORK

Initial work in link prediction often used **random negative sampling**, where negative samples were drawn from non-existent edges in the graph. This approach treats the problem as a binary classification task, distinguishing true links (positive samples) from non-links (negative samples)(Zhang & Chen, 2018b). However, randomly chosen negatives tend to be easy to classify, which can lead to suboptimal models that fail to generalize well, especially in large and sparse graphs(Wang et al., 2017b). More recent approaches have moved toward **dynamic negative sampling** and **hard negative mining** to address these limitations. Hard negatives are edges that are not present in the graph but share structural or feature similarities with positive edges. Studies have shown that selecting harder negative samples during training forces the model to learn more discriminative features, improving its ability to predict challenging or ambiguous links (Grover & Leskovec, 2016). Techniques such as **self-contrastive learning** have also been used to approximate the full negative set by leveraging negative edges dynamically during training, significantly enhancing performance in large-scale graphs (Wu & Zhu, 2019).

A parallel line of research has focused on using **labeling tricks** and incorporating **structural features** to address inherent limitations in graph-based models, such as automorphic node symmetry, where structurally similar nodes receive identical embeddings (Zhang et al., 2021; Zhu et al., 2021; Chamberlain et al., 2023). These methods, which explicitly add structural information, help models break symmetry and distinguish between nodes that may otherwise appear similar in purely topological terms (Chen & Liu, 2017). Negative sampling plays an important role in this context—when negatives are sampled from structurally similar nodes, the model is better trained to capture nuanced structural differences, thus improving link prediction performance.

Active Learning represents another related research direction, as it seeks to select a set of data for further labeling (Ren et al., 2021). RANS can be connected to active learning if we consider the special case where $f^{(i)}$ represents both the data-labeling and training models (Ren et al., 2021). Selecting the highest scoring elements of $\hat{\mathcal{D}}^{n'}$ under $f^{(i)}$ amounts to an Expected Error Reduction

process (Mussmann et al., 2022). However, RANS differs from active learning because it does not select datapoints for further labeling, as these labels are available trivially.

Another emerging area involves utilizing Generative Adversarial Networks (GANs) (Goodfellow et al., 2014) for hard-negative mining. Cai & Wang (2018) applies adversarial learning to improve the quality of negative sampling on knowledge graph embeddings. A separate knowledge graph embedding model (the generator) that produces hard negatives is trained alongside a model (the discriminator) that learns to distinguish between true and false triples on the graph. A similar approach is evaluated for information retrieval by Wang et al. (2017a), where the generator feeds relevant documents for a given query into the discriminator. Additionally, Yu et al. (2018) proposes a general framework for using GANs in graph representation learning tasks. While using the GANs framework for negative sampling allows for iterative learning of sampling and classification during training, promising high-quality negatives, the iterative nature can cause training instability and increased computational complexity. Diffusion-model based methods also provide a promising path forward towards improving the quality of the generated negative samples, but are computationally expensive (Nguyen & Fang, 2024). RANS can be viewed as a GAN without the minimax game, offering flexibility in when new samples are generated.

Several studies have indicated that the overall quality of negative sampling can significantly impact the expressiveness of link prediction models (Yang et al., 2020; Li et al., 2023; Yang et al., 2024). For instance, while Graph Neural Networks (GNNs) are often employed for their ability to learn rich node representations, their performance can be substantially enhanced by carefully curating negative samples during training (Zhang & Chen, 2016). Hard-negative sampling techniques have been shown to eliminate problems such as oversmoothing in GNNs, where node embeddings become too similar, by encouraging the model to differentiate between subtle structural or feature-based patterns (Yang et al., 2016a).

In conclusion, while much of the link prediction research has focused on improving model architectures, recent work highlights the critical role of negative sampling strategies. By shifting from random to more informed, structure-aware sampling methods, researchers have demonstrated substantial improvements in predictive accuracy. This paper extends these insights by investigating the impact of negative sampling on model performance and proposing a refined dynamic hard-negative sampling approach to optimize link prediction tasks.

## 3 BACKGROUND

**Graphs.** A graph $\mathcal{G} = (\mathcal{V}, \mathcal{E}, \mathbf{X})$ is a mathematical object where vertices $\mathcal{V}$ represent entities, and edges $\mathcal{E}$ capture the relationships between these entities. The vertex set $\mathcal{V}$ contains $N$ vertices, and the feature matrix $\mathbf{X} \in \mathbb{R}^{N \times d_0}$ represents $d_0$-dimensional features for each vertex. The graph can also be described by an adjacency matrix $\mathbf{A} \in \mathbb{R}^{N \times N}$, where $\mathbf{A}_{ij} = 1$ if there is an edge between vertices $i$ and $j$. Given $\mathbf{A}$, the combinatorial Laplacian $\mathbf{L}$ is defined as $\mathbf{L} = \mathbf{D}^{-\frac{1}{2}} \tilde{\mathbf{A}} \mathbf{D}^{-\frac{1}{2}}$, where $\tilde{\mathbf{A}} = \mathbf{I} - \mathbf{A}$ and $\mathbf{D}$ is the diagonal degree matrix of $\tilde{\mathbf{A}}$.

**Message Passing Neural Networks (MPNNs).** MPNNs generalize the concept of spatial convolution to graphs, allowing hidden representations of vertices to be computed by aggregating features over multiple graph layers. Starting with a graph $\mathcal{G}$ with node features $\mathbf{X}$, following the framework of Gilmer et al. (2017), an MPNN updates a vertex's hidden representation through a *message function* and an *update function* denoted by $\psi_l(\cdot)$ and $\phi_l(\cdot)$, respectively. The message function $\psi_l : \mathbb{R}^{d_l^{\text{in}}} \to \mathbb{R}^{d_l^{\text{out}}}$ transforms messages between vertices, while the update function $\phi_l : \mathbb{R}^{d_l^{\text{in}}} \times \mathbb{R}^{d_l^{\text{out}}} \to \mathbb{R}^{d_{l+1}^{\text{in}}}$ combines the current vertex state and aggregated messages to compute the next layer's representation. The hidden representation for vertex $i$ at layer $l+1$ is given by:

$$h_i^{l+1} = \phi_l \left( h_i^l, \sum_j \mathbf{T}_{ij} \psi_l \left( h_i^l, h_j^l \right) \right), \tag{1}$$

where $\mathbf{T}$ is the *transition matrix*, which generalizes the adjacency matrix $\mathbf{A}$ to account for potential graph rewiring during training (Topping et al., 2021; Markovich, 2023; Gasteiger et al., 2019). In principle, there are many choices for both $\phi_l(\cdot)$ and $\psi_l(\cdot)$, a common practice is to employ a learnable Multi-layer Perceptron (MLP) with shared weights across all vertices. After message passing,

the hidden representations are passed through a task-specific readout function, which may include residual connections (Zhu et al., 2020).

**Link Prediction.** Link prediction aims to learn a function $f(\mathbf{h}_i, \mathbf{h}_j)$ that predicts the likelihood of an edge between vertices $i$ and $j$ based on their node representations $\mathbf{h}_i$ and $\mathbf{h}_j$. Typically, node representations are obtained using an MPNN, i.e. $\mathrm{MPNN}(i, \mathcal{G}) = \mathbf{h}_i, \forall i \in \mathcal{V}$. Putting it all together, link prediction seeks to learn a readout function $f(\mathrm{MPNN}(i, \mathcal{G}), \mathrm{MPNN}(j, \mathcal{G})) \to \mathbb{R}$, treating it as a binary classification task (Hasan & Zaki, 2011). Popular choices for $f$ include dot product (Trouillon et al., 2016), Hadamard product (Wang et al., 2022; Chamberlain et al., 2023), or MLP (Chamberlain et al., 2023). Since link prediction is framed as a binary classification task, both positive and negative samples are required for training. Positive samples correspond to existing edges, while negative samples are drawn from non-existent edges. Since the number of possible negative edges ($\mathcal{O}(N^2)$) far exceeds the number of actual edges ($\mathcal{O}(|\mathcal{E}|)$), it is necessary to sample a subset of negative edges to maintain dataset balance and computational efficiency (Bordes et al., 2013).

## 4 Mechanisms of Negative Sampling

**Negative Sampling and Self-Contrastive Methods.** Let $\mathcal{V}' = \mathcal{V} \setminus \{k : (i, k) \in \mathcal{E}\}$ denote the set of nodes that node $i \in \mathcal{V}$ has no connection with. From a self-contrastive perspective, the goal in link prediction is to learn a function parameterized by $\theta$, modeling the conditional probability of an edge between vertices $v_i$ and $v_j$ as follows:

$$p(v_i | v_j; \theta) = \frac{e^{f_\theta(v_i) \cdot f_\theta(v_j)}}{\sum_{k \in \mathcal{V}'} e^{f_\theta(v_i) \cdot f_\theta(v_k)}}. \tag{2}$$

This objective is intractable for large datasets, but following Mikolov et al. (2013), it can be approximated as:

$$\arg\max_\theta \sum_{(v_i, v_j) \in \mathcal{E}} \ln \sigma(f_\theta(v_i) \cdot f_\theta(v_j)) + \sum_{(v_i, v_k) \in \mathcal{E}'} \ln \sigma(-f_\theta(v_i) \cdot f_\theta(v_k)), \tag{3}$$

where $\mathcal{E}'$ is a set of sampled negative edges. Minimizing this simplified objective yields a model that approximates the joint distribution of $(v_i, v_j)$, rather than the conditional probability. This approach is self-contrastive because all edges absent in the adjacency matrix are treated as negatives.

**Limitations of MPNNs for Link Prediction.** MPNNs, which are equivalent to the Weisfeiler-Leman isomorphism test, face two key limitations in link prediction. First, they cannot count triangles, a critical structural feature (Tolmachev et al., 2021; Chen et al., 2020). Second, MPNNs suffer from the automorphic node problem, where they assign equivalent representations to vertices in the same graph orbit (i.e., under graph automorphisms) (Srinivasan & Ribeiro, 2019). This leads to two issues: (1) $f(v_i, v_j) = f(v_i, v_k)$ for vertices $v_j$ and $v_k$ in the same orbit, regardless of distance; and (2) when the MPNN suffers from oversmoothing, it produces overly similar vertex representations, making the link predictor too optimistic.

**Negative Sampling and Structural Representations.** To improve link prediction performance, researchers often employ techniques such as labeling tricks and explicit structural features (Zhang et al., 2021; Zhu et al., 2021; Chamberlain et al., 2023). These methods help break automorphic node symmetry by enabling the edge-wise decoder to learn how to incorporate these additional sources of information. Consider the naphthalene molecule shown in Figure 1, which contains five sets of structurally isomorphic vertices. Given that vertices $v_9$ and $v_{17}$ are isomorphic, any link predictor based purely on structural node representations will yield equivalent representations for edges $(v_1, v_9)$ and $(v_1, v_{17})$. The scenario becomes more nuanced when incorporating additional features, such as the number of valence electrons assigned to vertices (atoms) in the graph (molecule). In naphthalene, atoms in the central rings possess four valence electrons, while those on the periphery have one valence electron.

To address these issues, we can define a link predictor that incorporates both structural node representations and the *distance* between node pairs. While this approach can break symmetry, it is effective only when negative samples during training include pairs of structurally isomorphic vertices. For instance, if negatives are chosen solely from vertices that differ significantly in feature

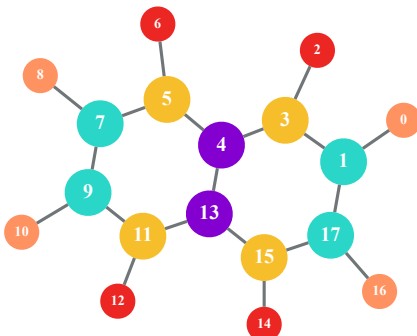

Figure 1: **Naphthalene molecule as a graph**. Structurally isomorphic atoms are coloured same, central ring atoms are represented larger than outer atoms in vertex size.

space, the model will experience weak optimization pressure to emphasize distance or other structural information, as feature information alone suffices to discriminate between positive and negative edges (Srinivasan & Ribeiro, 2019; Zhang et al., 2021). Conversely, when negatives with similar feature representations are chosen during training, the model experiences the necessary optimization pressure to focus on distance-based features. Thus, the quality of the link predictor depends not only on model expressivity but also on the negative samples observed during training (Yang et al., 2020; 2024). Importantly, negative sampling does not alter the expressivity of the hypothesis space; rather, it helps identify the optimal model within that space (Zhang et al., 2023).

We can formalize this argument by analyzing an analytically tractable model. Consider a 2-layer linearized GCN, where the functional form $f(v_i, v_j)$ is given by:

$$f_{\text{GCN}}(v_i, v_j) = (\mathbf{T}\Theta_2\mathbf{T}\Theta_1\mathbf{X})[:, i] \cdot (\mathbf{T}\Theta_2\boldsymbol{T}\Theta_1\mathbf{X})[:, j], \tag{4}$$

where $\Theta_1 \in \mathbb{R}^{d_0 \times d_1}$ and $\Theta_2 \in \mathbb{R}^{d_1 \times d_2}$ denote the parameters at update layer $l$. This formulation yields two key observations. First, graph connectivity is accounted for only in $\mathbf{T}^2$, with second-order sums being independent, causing the model to estimate high scores for edges with endpoints from disparate graph neighborhoods but similar feature distributions. This proves problematic for GNNs prone to oversmoothing, as node embeddings converge to similar values, making their dot products approach 1. Consequently, any two vertices with $f_{\text{GCN}}(v_i; \mathcal{G}) = f_{\text{GCN}}(v_j; \mathcal{G})$ will be labeled as structurally isomorphic, leading to identical results for edge queries. Second, this formulation constrains the link representation to a limited field of $l$, meaning higher-order graph structures such as triangle closures cannot be captured unless explicitly added through the readout function.

## 5 ANALYZING NEGATIVE SAMPLING AND EMPIRICAL RISK

An important question to explore is: *"How does the quality of negative samples affect the generalization performance of link prediction models?"*. To answer this, we can frame the link prediction task as a binary classification problem and analyze it using the tools of **empirical risk minimization**. We start by defining the **risk** $R(f)$ of a classifier $f$ as:

$$R(f) \triangleq \mathbb{E}_X \mathbb{E}_{Y|X}[\ell(Y, f(X))], \tag{5}$$

where $p(Y|X)$ represents the ground truth distribution of an edge's existence ($Y = 1$) given some features (such as node features, node identity, or edge features).

Assume that the model $f : \mathbb{R}^{\mathcal{D}} \to [0, 1]$ belongs to a class of models $\mathcal{F}$, and each $f$ is parameterized by $\theta$, representing a distinct model within $\mathcal{F}$. The loss function $\ell(f(x_{ij}), y_{ij}) : \mathbb{R}^{\mathcal{D}} \to \mathbb{R}^+$ quantifies prediction error.

Given access to samples from $p(Y|X)$, we construct the **empirical risk** based on a dataset $\mathcal{D} = \{x_k, y_k\}_{k=1}^{K}$, under the assumption of $\mathcal{D} \overset{\text{iid}}{\sim} p(Y, X)$. In this particular setting of edge prediction, observing a given graph can be viewed as such an iid dataset. For a sampled graph, the empirical

risk is defined as:

$$\tilde{R}(f) \triangleq \frac{1}{|\mathcal{D}|} \sum_{k=1}^{|\mathcal{D}|} \ell(y_k, f(x_k)) \quad \text{with} \quad x_k, y_k \sim p(X, Y). \tag{6}$$

As it is a proper empirical risk, $\tilde{R}$ enjoys the usual guarantees and bounds with respect to the true risk $R(f)$. However, calculating the full empirical risk is computationally expensive, as it involves iterating through all possible node pairs. Instead, we rely on a subset of samples $\hat{\mathcal{D}} \subset \mathcal{D}$ to compute an approximate empirical risk:

$$\hat{R}(f) \triangleq \frac{1}{|\hat{\mathcal{D}}|} \sum_{(x_k, y_k) \in \hat{\mathcal{D}}} \ell(f_\theta(x_k), y_k). \tag{7}$$

We aim to minimize the **true risk** $R(f)$, but since we only work with sampled data, we instead minimize the empirical risks:

$$f^* \triangleq \arg\min_{f \in \mathcal{F}} R(f), \quad \tilde{f}^* \triangleq \arg\min_{f \in \mathcal{F}} \tilde{R}(f), \quad \hat{f}^* \triangleq \arg\min_{f \in \mathcal{F}} \hat{R}(f). \tag{8}$$

Let $\mathcal{D}^p$ and $\hat{\mathcal{D}}^p$ ($\mathcal{D}^n$ and $\hat{\mathcal{D}}^n$) denote the positives (negatives) in $\mathcal{D}$ and $\hat{\mathcal{D}}$, respectively and define the sum of losses over positive and negative samples for a given classifier $f$ as follows:

$$\ell_f^+ = \sum_{(x_k, y_k) \in \mathcal{D}^p} \ell(y_k, f(x_k)), \quad \ell_f^- = \sum_{(x_k, y_k) \in \mathcal{D}^n} \ell(y_k, f(x_k)), \tag{9}$$

$$\hat{\ell}_f^+ = \sum_{(x_k, y_k) \in \hat{\mathcal{D}}^p} \ell(y_k, f(x_k)), \quad \hat{\ell}_f^- = \sum_{(x_k, y_k) \in \hat{\mathcal{D}}^n} \ell(y_k, f(x_k)). \tag{10}$$

Now, let's examine the gap between the risks $\tilde{R}$ and $\hat{R}$ for a classifier $f$.

$$|\tilde{R}(f) - \hat{R}(f)| = \left| \frac{\ell_f^+ + \ell_f^-}{|\mathcal{D}|} - \frac{\hat{\ell}_f^+ + \hat{\ell}_f^-}{|\hat{\mathcal{D}}|} \right|. \tag{11}$$

Assuming that the set of positives is the same $\mathcal{D}^p \approx \hat{\mathcal{D}}^p$, and defining $m_f^-$ as **the risk associated with negative edges** in $\mathcal{D}/\hat{\mathcal{D}}$, yields $\ell_f^- = m_f^- + \hat{\ell}_f^-$, and we arrive at Theorem 5.1.

**Theorem 5.1** (Link Prediction Generalization Gap). *Using the definitions for $\ell^\pm$ and $m_f^-$ above, the generalization gap is:*

$$|\tilde{R}_f - \hat{R}_f| = \frac{1}{|\mathcal{D}|} \left| \left(1 - \frac{|\mathcal{D}|}{|\hat{\mathcal{D}}|}\right) \hat{R}_f + m_f^- \right|. \tag{12}$$

The term $m_f^-$ can be interpreted as the contribution from the missing negative edges from $\hat{D}$, and is always positive. The term $\left(1 - \frac{|\mathcal{D}|}{|\hat{\mathcal{D}}|}\right) \hat{R}_f$ is the empirical risk on the subset $\hat{\mathcal{D}}$ scaled by the factor $\left(1 - \frac{|\mathcal{D}|}{|\hat{\mathcal{D}}|}\right)$, which is always negative as $|\mathcal{D}| > |\hat{\mathcal{D}}|$. Many real world graphs are sparse, meaning that $|\hat{\mathcal{D}}_{p,n}| \approx N$, which allows us to further simplify the first term to $(1 - N) \hat{R}_f$.

While the generalization gap is interesting for any element of $\mathcal{F}$, we are usually interested in $\hat{f}^*$, which is found through some training procedure like Stochastic Gradient Descent (SGD) (Robbins & Monro, 1951). Given that $\hat{f}^*$ is the minimizer of $\hat{R}$, the first term should take its minimum value as well. The $m_f^-$ term, however, is not minimized by $\hat{f}^*$. Furthermore, while the samples that comprise $\hat{\mathcal{D}}$ are drawn uniformly, the distribution of our loss values will not be similarly uniform. Given that $\hat{R}_{f^*}$ is at its minimum, the minimum generalization gap occurs when $\hat{R}_{f^*} = m_{f^*}^-$, but

we note that if there are $N$ terms on the left, there are $N^2$ terms on the right side of equation. This naturally implies that shrinking our generalization gap requires carefully choosing the hardest negative edges for $\hat{f}^*$ given $\hat{\mathcal{D}}$. Thus, we must carefully provide a representative sample of the loss distribution. This problem is particularly pernicious when we perform our minimization with SGD, because SGD minimizes the loss, and even hard edges for model $\hat{f}^n$ will become easier for model $\hat{f}^{n+1}$, where the superscripts indicate the SGD step index. Even though $\hat{R}_{f^n} > \hat{R}_{f^{n+1}}$, we have no guarantee that $m_{f^n}^- > m_{f^{n+1}}^-$. As a result, minimization of $\hat{R}$ provides an overly optimistic estimate for the performance of $\hat{f}^*$ unless negatives are chosen carefully.

With the generalization gap in hand, it is next interesting to bound the excess risk that $f$ incurs when compared to $f^*$, which we do through application of Theorem 5.1

**Theorem 5.2** (Link Prediction Excess Risk). *Excess risk for our empirical risk minimizer, $\hat{f}^*$, is equal to:*

$$\mathcal{E}(\hat{f}^*) = \left| \tilde{R}_{\tilde{f}^*} - \hat{R}_{\hat{f}^*} \right| \leq 2 \frac{1}{|\mathcal{D}|} \sup_{f \in F} \left| m_f^- \right|. \tag{13}$$

*The proof of this theorem is given in Appendix 10.1.*

Traditionally, excess risk characterizes the discrepancy between a model $\hat{f}$ and $\tilde{f}^*$ over $\mathcal{F}$. In this view, increasing the expressivity of $\mathcal{F}$ can lead to an overall reduction in excess risk by enabling better data representations. We propose an alternative perspective: instead of modifying $\mathcal{F}$, we can reduce excess risk by exploring different data splitting strategies while holding $\mathcal{F}$ constant. If we construct $\mathcal{D}^n$ to contain only difficult edges under the model $\tilde{f}^*$, the second term will shrink because the remaining edges are "easy." Consequently, the selection of negative samples can significantly influence our choice of $\hat{f}^*$, thereby reducing both our uniform error bound and excess risk.

# 6 RISK AWARE NEGATIVE SAMPLING

We now develop a negative sampling algorithm that leverages these insights to generate better negative samples during training. Direct application of Theorem 5.2 would require both enumeration of all possible negative samples and access to $\tilde{f}^*$. However, if we had access to $\tilde{f}^*$, training would be unnecessary since we would already possess the optimal link predictor. In such a case, the optimal algorithm would simply return $\tilde{f}^*$.

---

**Algorithm 1** Risk Aware Negative Sampling

---

**Input:** $f, \hat{\mathcal{D}}, \delta, \eta$
  1: $\mathbf{Z} = \text{MPNN}(\mathcal{G}, \mathbf{X})$
  2: $p = f(\hat{\mathcal{D}}, \mathbf{Z})$                                        ▷ Score all training edges
  3: $N_{\text{easy}} = \sum_{\text{negs}} \mathbb{I}(p < \delta)$
  4: **if** $N_{\text{easy}} > \eta|\hat{\mathcal{D}}^n|$ **then**
  5:     $\hat{\mathcal{D}}^{n'} \sim q(\mathcal{G})$ s.t. $|\hat{\mathcal{D}}^{n'}| = kN_{easy}$     ▷ Sample new negatives from the base distribution, $q$.
  6:     $p = f(\hat{\mathcal{D}}^{n'}, \mathbf{Z})$                              ▷ Score all new negative edges
  7:     $\hat{\mathcal{D}}_{easy}^n = \hat{\mathcal{D}}_{hardest}^{n'}$
  8: **end if**

---

To circumvent this problem, we make several key observations. First, at the $i^{\text{th}}$ epoch, our most principled estimate of $\tilde{f}^*$ is $f^{(i)}$. Second, our generalization gap is governed by both $\hat{R}$, which we actively minimize, and $m_f^-$, which remains unknown. Third, we retain the freedom to reconstruct $\hat{\mathcal{D}}$ as needed. These observations lead to a straightforward conclusion: at epoch $i$, we should replace "easy" negatives with harder ones.

We accomplish this by sampling a set of new negatives according to a base distribution $q$, such that $\hat{\mathcal{D}}^{n'} = \{e \sim q(\mathcal{G})\}$. This approach leads to our method, Risk Aware Negative Sampling (RANS), defined in Algorithm 1. RANS first tests for the percentage of edges that the current model finds "easy." When this percentage exceeds a prescribed threshold, RANS randomly samples new

Table 1: **Principal comparison.** The prediction performance of GCN combined with different sampling techniques across different datasets. PNS results are omitted for OGBL-DDI because this dataset has no node features.

| Metric | CORA HR@10 | CITESEER HR@10 | PUBMED HR@100 | CHAMELEON HR@10 | SQUIRREL HR@100 | OGBL-DDI HR@20 | OGBL-COLLAB HR@50 |
|---|---|---|---|---|---|---|---|
| UNS-S | $26.28_{\pm 4.52}$ | $25.54_{\pm 5.49}$ | $18.74_{\pm 1.48}$ | $20.56_{\pm 7.04}$ | $46.70_{\pm 2.66}$ | $36.82_{\pm 4.23}$ | $42.10_{\pm 2.06}$ |
| USNS-S | $36.75_{\pm 4.21}$ | $28.07_{\pm 5.64}$ | $22.83_{\pm 3.77}$ | $27.42_{\pm 5.45}$ | $41.72_{\pm 2.17}$ | $36.97_{\pm 5.71}$ | $42.10_{\pm 1.55}$ |
| RWNS-S | $40.01_{\pm 6.78}$ | $48.92_{\pm 2.02}$ | $27.37_{\pm 3.68}$ | $14.34_{\pm 9.37}$ | $1.38_{\pm 1.06}$ | $33.07_{\pm 6.27}$ | $42.26_{\pm 1.62}$ |
| PNS-S | $22.33_{\pm 6.78}$ | $9.89_{\pm 3.35}$ | $15.88_{\pm 4.32}$ | $18.28_{\pm 6.15}$ | $33.16_{\pm 1.98}$ | - | OOM |
| UNS-D | $43.22_{\pm 4.80}$ | $54.22_{\pm 4.21}$ | $27.80_{\pm 2.13}$ | $21.64_{\pm 7.62}$ | $\mathbf{47.71}_{\pm 3.05}$ | $41.08_{\pm 7.48}$ | $44.75_{\pm 1.07}$ |
| USNS-D | $47.72_{\pm 4.21}$ | $58.07_{\pm 5.88}$ | $27.56_{\pm 2.51}$ | $12.52_{\pm 6.27}$ | $21.38_{\pm 6.36}$ | $23.56_{\pm 3.97}$ | $41.24_{\pm 1.79}$ |
| RWNS-D | $45.93_{\pm 6.52}$ | $47.34_{\pm 4.04}$ | $24.40_{\pm 0.95}$ | $8.96_{\pm 4.99}$ | $22.70_{\pm 1.31}$ | $19.53_{\pm 4.08}$ | $38.10_{\pm 0.78}$ |
| MCNS-D | $49.40_{\pm 4.14}$ | $55.98_{\pm 3.30}$ | $28.26_{\pm 2.37}$ | $7.89_{\pm 1.94}$ | $18.42_{\pm 3.88}$ | $36.40_{\pm 6.57}$ | $43.06_{\pm 1.30}$ |
| RANS | $\mathbf{51.36}_{\pm 3.64}$ | $\mathbf{61.01}_{\pm 1.21}$ | $\mathbf{33.17}_{\pm 3.38}$ | $\mathbf{35.50}_{\pm 6.16}$ | $45.81_{\pm 3.44}$ | $\mathbf{47.96}_{\pm 5.12}$ | $\mathbf{47.47}_{\pm 1.10}$ |

negatives according to base distribution $q$ and scores them using the current model. It then replaces the easiest negatives in the current dataset with the hardest new negatives. Our algorithm depends on three hyperparameters: $\delta$, $\eta$, and $k$. Here, $\delta$ defines the threshold for considering a negative edge "easy," $\eta$ specifies the proportion of edges that must be easy before regeneration, and $k$ determines the oversampling ratio. In our experiments, we set $\delta = 0.1\bar{p}_p$, where $\bar{p}_p$ is the average score of positive edges; $\eta = 0.95$; and $k = 10$, though $k = 2$ suffices for OGBL datasets. RANS also requires a base distribution $q$ from which we can efficiently generate many samples. In practice, we sample from the uniform distribution for computational efficiency, though any base distribution would suffice. Notably, the number of negatives is controllable, providing a convenient way to balance computational expense and negative mining.

## 7 EXPERIMENTS

In this section, we first describe for the experimental setup and we present an evaluation of our proposed negative sampling technique, comparing its performance against a range of established methods across multiple datasets. Next, we explore the generalization gap, model complexity, and robustness of different sampling techniques under varying conditions through a set of sensitivity analysis.

### 7.1 PRINCIPAL COMPARISON

**Setup.** To isolate the effects of negative sampling on model performance, we maintain a controlled experimental setup with fixed model architecture and hyperparameters. We employ a GCN as our base model for learning vertex embeddings. These embeddings are combined using the Hadamard product and passed through an MLP. Hyperparameters for both the encoder and edge predictor are detailed in Table 10.2 in the appendix. All experiments were conducted using PYTORCH GEOMETRIC 2.6.0 and PYTORCH 2.4 on an NVIDIA DGX A100 system with 128 AMD ROME 7742 cores and 8 NVIDIA A100 GPUs.

**Datasets.** We evaluate our method on seven datasets: CORA, CITESEER, and PUBMED from the Planetoid datasets (Yang et al., 2016b); CHAMELEON and SQUIRREL from the WebKB dataset (Pei et al., 2020); and OGBL-DDI and OGBL-COLLAB (Hu et al., 2020). For the Planetoid and WebKB datasets, we generate five different splits used across all experiments. For the OGBL datasets, we perform ten repetitions following the default experimental setup provided by the OGBL team. Results are averaged over all splits, with both mean and standard deviation reported.

**Metrics.** We use *hit rate* (HR) as our evaluation metric, a standard measure in link prediction tasks. HR measures the proportion of true positive edges ranked within the top $k$ predictions, defined as:

$$\text{HR}@k = \frac{\text{Number of true positives in top } k}{\text{Total number of true positives}}, \tag{14}$$

where $k$ is a user-defined threshold, and a "hit" occurs when a true positive edge (i.e., a correct link between two nodes) ranks among the top $k$ predicted links. We evaluate prediction performance using HR@10, HR@20, HR@50, and HR@100, specifying the metric used in each case.

**Baselines.** We compare our proposed negative sampling method against several established techniques:

- *UNS: Uniform Negative Sampling* selects negative samples uniformly at random from all possible edges. This is the default negative sampling technique in PyTorch Geometric and does not consider structural or feature-based information when selecting negatives (Bordes et al., 2013; Fey & Lenssen, 2019).

- *USNS: Uniform Structural Negative Sampling* samples negatives uniformly from nodes sharing structural similarity. This approach generates more challenging negative samples compared to UNS by focusing on structurally similar nodes (Wang et al., 2014; Fey & Lenssen, 2019).

- *RWNS: Random Walk Weighted Negative Sampling* selects negative samples based on random walks over the graph. Selection probability is weighted by node frequency in random walks, reflecting connectivity and structural importance (Hamilton et al., 2017).

- *PNS: Personalized Negative Sampling* selects negative samples based on node feature similarity, with selection probability increasing as feature similarity increases (Pal et al., 2020).

- *MCNS: Monte Carlo Negative Sampling* employs Monte Carlo methods for negative sampling, repeatedly sampling possible negatives and estimating their loss contribution using probabilistic techniques (Yang et al., 2020).

These methods operate in either static or dynamic settings. In static settings, negative training edges are generated once and used throughout all epochs. In dynamic settings, edges are regenerated every $K$ epochs, where $K$ is a tuned hyperparameter. UNS, USNS, and RWNS support both settings, denoted by -S (static) or -D (dynamic) suffixes. PNS is limited to static settings due to computational constraints, while MCNS is inherently dynamic. We omit PNS for OGBL-DDI due to the absence of features and for OGBL-COLLAB due to memory constraints.

Table 6 presents our experimental results. RANS consistently improves the accuracy of our base GCN across all datasets compared to other negative sampling techniques, validating our hypothesis that risk-aware sampling advances the performance of graph learning methods. We provide a table comparing per-epoch average run-times in Table 5, and observe only modest increases in computation expense with RANS when compared with the baselines. In Figure 3 we report the hyperparameter sensitivity to $\delta$ and $\eta$, and observe that there is a large region where RANS improves performance. This leads us to conclude that RANS is robust to these hyperparameters, and that extensive tuning is not warranted.

**Generalization Gap.** A common observation in link prediction tasks is that training accuracy converges faster than validation and test accuracy, empirically demonstrating the generalization gap. Theorem 5.1 and our subsequent analysis indicate that minimizing $m_f^-$ can reduce this gap. As shown in Figure 2, RANS delays the saturation of training accuracy by introducing challenging new negatives that temporarily decrease training accuracy. This approach results in a significantly smaller train-test generalization gap for GCN with RANS compared to UNS-S, the standard negative sampling strategy in link prediction training.

Table 2: **Distribution-swap on negatives.** The predictive performance (HR@20) of different configurations for the distribution of negative sampling on train and test split. All experiments were run on the Cora dataset.

| Train \ Test | UNS | USNS | PNS |
|---|---|---|---|
| UNS-S | $50.2_{\pm 4.8}$ | $41.8_{\pm 3.7}$ | $13.0_{\pm 2.8}$ |
| USNS-S | $39.5_{\pm 4.1}$ | $40.2_{\pm 3.2}$ | $13.6_{\pm 3.1}$ |
| PNS | $35.0_{\pm 3.2}$ | $31.4_{\pm 3.6}$ | $21.5_{\pm 1.9}$ |
| RANS | $59.3_{\pm 2.6}$ | $69.8_{\pm 4.3}$ | $22.7_{\pm 2.6}$ |

**Generalization Gap Across Distributions.** Another approach to exploring the generalization gap is to examine settings where train and test negatives are drawn from different distributions. This scenario corresponds to industrial applications with nonstationary negative distributions (Zhang et al., 2016; Ma et al., 2007) or cold-start settings (Wei et al., 2021; Du et al., 2022). Theorem 5.1 indicates that for a model $\hat{f}^*$ that minimizes $\hat{R}$, generalization performance is largely governed by

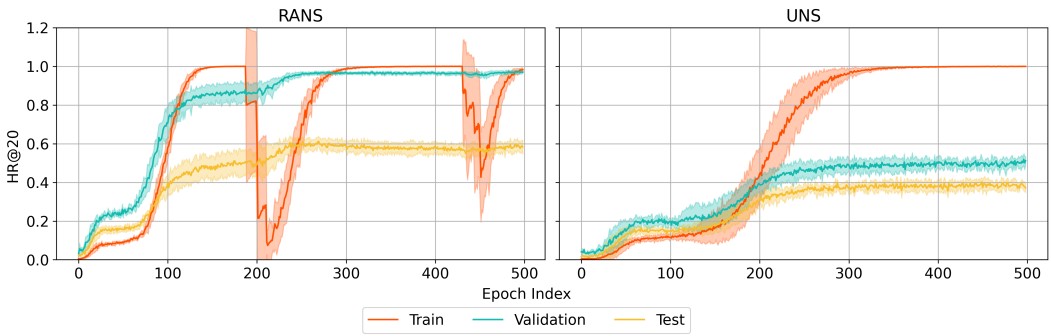

Figure 2: **Generalization gap.** Comparison of predictive performance (HR@20) between GCN with RANS (left) and GCN with uniform negative sampling (UNS) (right) across epochs on training, validation, and test splits.

$m_f^-$. To empirically validate this effect, we designed experiments where models were trained using one negative sampling strategy and tested using negatives generated by a different strategy. All experiments maintained consistent model architecture and were averaged over five data splits, with identical splits used across experiments. Table 2 presents the results of these **distribution-swap on negatives** experiments. Models trained with fixed negative sampling strategies perform poorly on test sets with substantially different negative distributions, a trend consistent across all standard negative sampling approaches. In contrast, RANS achieves superior generalization performance across all test sets.

Table 3: **SOTA Comparison**. The predictive performance of different models in comparison to GCN+RANS.

| Metric | CORA HR@100 | CITESEER HR@100 |
|---|---|---|
| Neo-GNN | $80.42_{\pm 1.31}$ | $84.67_{\pm 2.16}$ |
| SEAL | $81.12_{\pm 1.84}$ | $86.32_{\pm 1.59}$ |
| ELPH | $87.72_{\pm 2.13}$ | $93.44_{\pm 0.53}$ |
| BUDDY | $88.00_{\pm 0.44}$ | $92.93_{\pm 0.27}$ |
| GCN+UNS | $66.79_{\pm 1.65}$ | $67.08_{\pm 2.94}$ |
| GCN+RANS | $80.06_{\pm 2.38}$ | $88.16_{\pm 3.03}$ |
| HLGNN+UNS | $88.96_{\pm 2.17}$ | $93.01_{\pm 2.71}$ |
| HLGNN+RANS | $\mathbf{89.63}_{\pm \mathbf{2.10}}$ | $\mathbf{95.21}_{\pm \mathbf{1.35}}$ |

**Model Complexity vs. Negative Quality** Following insights from Theorem 5.2, we investigate how a simple model like our GCN, when augmented with RANS, compares to state-of-the-art models. This investigation is significant because model underperformance has traditionally been attributed to limited hypothesis space expressiveness. In Table 3, we compare the performance of our untuned GCN+RANS against leading methods including Neo-GNN (Yun et al., 2021), SEAL (Zhang & Chen, 2018b), and BUDDY (Chamberlain et al., 2023). GCN+RANS achieves comparable performance to both SEAL and Neo-GNN despite operating in a significantly less expressive hypothesis space. While BUDDY maintains superior performance, RANS substantially narrows this performance gap. All performance numbers, except for GCN+RANS reported in Table 3 are taken from the literaure (Chamberlain et al., 2023), so as to capture optimal performance after extensive hyperparameter tuning so as not to unfairly privilege RANS. Importantly, RANS serves as a training augmentation applicable to any model, suggesting potential performance improvements across all link prediction models when combined with this approach.

# 8 CONCLUSION

In conclusion, we have presented a new technique for generating negative samples dynamically that leads to higher quality models. This approach is inspired by an analysis of the excess risk in link prediction, and we show both theoretically and empirically that our method reduces the generalization gap, and leads to better model performance. This work is limited to static link prediction contexts, but future work will explore temporal graphs.

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

# 9 APPENDIX

# 10 PROOFS

## 10.1 PROOF OF THEOREM 4.2

To complete this proof, we first need to compute the uniform error bound.

$$\epsilon = \sup_{f \in \mathcal{F}} \left| \hat{R}_f - \tilde{R}_f \right| \tag{15}$$

$$\epsilon = \sup_{f \in \mathcal{F}} \left| \frac{\hat{\ell}_f^+ + \hat{\ell}_f^-}{\left| \hat{\mathcal{D}} \right|} - \frac{\ell_f^+ + \ell_f^-}{|\mathcal{D}|} \right| \tag{16}$$

$$\epsilon = \sup_{f \in \mathcal{F}} \left| \frac{\hat{\ell}_f^+ + \hat{\ell}_f^-}{\left| \hat{\mathcal{D}} \right|} - \frac{\hat{\ell}_f^+ + \hat{\ell}_f^- + m_f^-}{|\mathcal{D}|} \right| \tag{17}$$

$$\epsilon = \sup_{f \in \mathcal{F}} \left| \frac{\hat{R}_f}{\left| \hat{\mathcal{D}} \right|} - \frac{\hat{F}_f}{|\mathcal{D}|} - \frac{m_f^-}{|\mathcal{D}|} \right| \tag{18}$$

$$\epsilon = \sup_{f \in \mathcal{F}} \frac{1}{|\mathcal{D}|} \left| \hat{R}_f \left( \frac{|\mathcal{D}|}{\left| \hat{\mathcal{D}} \right|} - 1 \right) - m_f^- \right| \tag{19}$$

Supremum occurs when $\hat{R}_f$ is minimized while $m_f^-$ is maximized, because $m_f^-$ has $N^2$ terms while $\hat{R}$ only has $N$.

$$\epsilon \leq \sup_{f \in \mathcal{F}} \frac{1}{|\mathcal{D}|} \left| m_f^- \right| \tag{20}$$

with this in hand, we want to estimate

$$\mathcal{E}(\hat{f}^*) = \tilde{R}(\hat{f}^*) - \tilde{(R)}(\tilde{f}^*) \tag{21}$$

First,

$$\hat{R}(\hat{f}^*) \leq \hat{R}(f) \ \ \forall f \in \mathcal{F}, \tag{22}$$

then, using the uniform error bound,

$$\left| \hat{R}(\hat{f}^*) - \tilde{R}(\hat{f}^*) \right| \leq \epsilon \tag{23}$$

and

$$\left| \hat{R}(\tilde{f}^*) - \tilde{R}(\tilde{f}^*) \right| \leq \epsilon \tag{24}$$

implies that

$$R(\hat{f}^*) \leq \hat{R}(\hat{f}^*) + \epsilon \tag{25}$$

and

$$R(\tilde{f}^*) \leq \hat{R}(\tilde{f}^*) + \epsilon. \tag{26}$$

Starting from:

$$R(\hat{f}^*) \leq \hat{R}(\hat{f}^*) + \epsilon \tag{27}$$

we can use the fact that $\hat{R}(\hat{f}^*) \leq \hat{R}(\tilde{f}^*)$ to get

$$R(\hat{f}^*) \leq \hat{R}(\tilde{f}^*) + \epsilon. \tag{28}$$

Using that $R(\tilde{f}^*) \leq \hat{R}(\tilde{f}^*) + \epsilon$, we find that

$$R(\hat{f}^*) \leq \left( R(\tilde{f}^*) \leq \hat{R}(\tilde{f}^*) + \epsilon \right) + \epsilon \tag{29}$$

which yields:

$$R(\hat{f}^*) \leq R(\tilde{f}^*) \leq \hat{R}(\tilde{f}^*) + 2\epsilon \tag{30}$$

$$R(\hat{f}^*) \leq R(\tilde{f}^*) - \hat{R}(\tilde{f}^*) \leq 2\epsilon \tag{31}$$

Returning to our excess risk definition:

$$\mathcal{E}(\hat{f}^*) = \tilde{R}(\hat{f}^*) - \tilde{(R)}(\tilde{f}^*) \tag{32}$$

We find that:

$$\mathcal{E}(\hat{f}^*) \leq 2\epsilon \leq 2 \frac{1}{|\mathcal{D}|} \sup_{f \in F} \left| m_f^- \right| \tag{33}$$

## 10.2 ENCODER AND DECODER PARAMETERS

| Parameter | Value |
|---|---|
| Encoder Number of Layers | 2 |
| Encoder Hidden Dimension | 256 |
| Encoder Output Dimension | 256 |
| Encoder Dropout | 0.2 |
| Decoder Num Layers | 2 |
| Decoder Hidden Dimension | 256 |
| Decoder Output Dimension | 256 |
| Decoder Dropout | 0.3 |
| Learning Rate | $1.13 \times 10^{-4}$ |
| Weight Decay | $8.6 \times 10^{-4}$ |

Table 4: Model Parameters

## 10.3 RUN TIME COMPARISONS

## 10.4 HYPERPARAMETER SENSITIVITY

## 10.5 PRINCIPAL COMPARISON - HLGNN

|        | CORA | CITESEER | PUBMED | CHAMELEON | SQUIRREL | OGBL-DDI | OGBL-COLLAB |
|--------|------|----------|--------|-----------|----------|----------|-------------|
| UNS-S  | 56s  | 38s      | 95s    | 56s       | 284s     | 5442s    | 5295s       |
| USNS-S | 56s  | 56s      | 531s   | 163s      | 272s     | 1482s    | 5308s       |
| RWNS-S | 35s  | 34s      | 157s   | 95s       | 313s     | 1469s    | 5729s       |
| PNS-S  | 93s  | 93s      | 170s   | 987s      | 192s     | -        | OOM         |
| UNS-D  | 30s  | 29s      | 101s   | 46s       | 290s     | 5289s    | 6563s       |
| USNS-D | 31s  | 29s      | 99s    | 57s       | 284s     | 72553s   | 6142s       |
| RWNS-D | 35s  | 33s      | 156s   | 94s       | 321s     | 36939s   | 5491s       |
| MCNS-D | 31s  | 29s      | 99s    | 57s       | 287s     | 18611s   | 9876s       |
| RANS   | 31s  | 31s      | 100s   | 62s       | 288s     | 27791s   | 11502s      |

Table 5: Observed model runtimes.

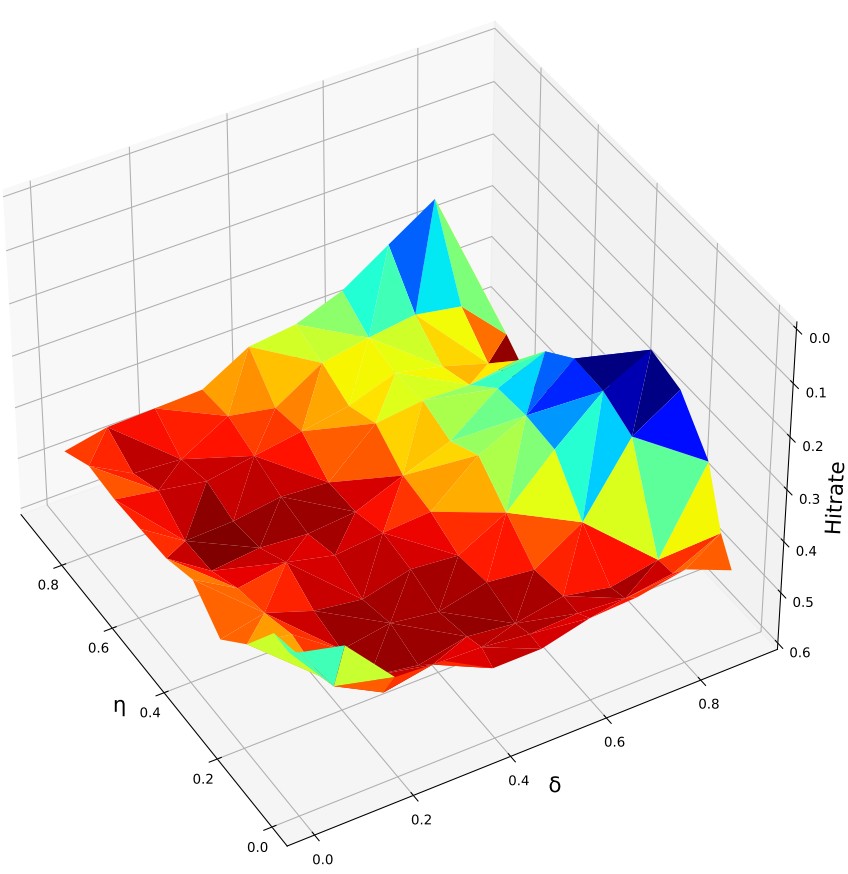

Figure 3: **Hyperparameter Sensitivity** The hyperparameter sensitivity for GCN+RANS on the Cora dataset. We have computed the HR@10 of GCN+RANS for $\eta$ and $\delta$ and plot the surface. We observe that there is a broad region of reasoanble performance corresponding to the red basin.

Table 6: **Principal comparison.** The prediction performance of HLGNN combined with different sampling techniques across different datasets.

| Metric | CORA HR@10 | CITESEER HR@10 | PUBMED HR@100 | CHAMELEON HR@10 | SQUIRREL HR@10 | OGBL-COLLAB HR@50 |
|---|---|---|---|---|---|---|
| UNS-S | $64.67_{\pm 6.34}$ | $75.69_{\pm 1.54}$ | $86.86_{\pm 0.86}$ | $49.98_{\pm 6.76}$ | $81.53_{\pm 0.96}$ | $47.31_{\pm 1.92}$ |
| USNS-S | $67.40_{\pm 3.51}$ | $73.10_{\pm 3.88}$ | $84.94_{\pm 0.95}$ | $49.62_{\pm 9.83}$ | $71.71_{\pm 3.30}$ | $47.20_{\pm 2.78}$ |
| RWNS-S | $67.97_{\pm 3.57}$ | $77.58_{\pm 1.65}$ | $84.17_{\pm 1.09}$ | $27.40_{\pm 4.61}$ | $18.73_{\pm 1.85}$ | $41.56_{\pm 1.20}$ |
| UNS-D | $63.23_{\pm 6.52}$ | $75.30_{\pm 3.52}$ | $86.87_{\pm 0.79}$ | $58.49_{\pm 8.42}$ | $\mathbf{83.08}_{\pm 2.66}$ | $49.45_{\pm 2.29}$ |
| USNS-D | $63.07_{\pm 2.74}$ | $75.30_{\pm 3.51}$ | $84.73_{\pm 0.76}$ | $62.12_{\pm 3.63}$ | $81.53_{\pm 0.96}$ | $48.70_{\pm 0.80}$ |
| RWNS-D | $64.52_{\pm 1.85}$ | $75.91_{\pm 1.31}$ | $84.81_{\pm 0.72}$ | $31.84_{\pm 9.24}$ | $38.46_{\pm 10.06}$ | $41.84_{\pm 1.22}$ |
| MCNS-D | $59.35_{\pm 3.67}$ | $76.00_{\pm 2.03}$ | $85.14_{\pm 0.75}$ | $44.27_{\pm 9.74}$ | $75.51_{\pm 1.95}$ | $47.77_{\pm 1.69}$ |
| RANS | $\mathbf{68.81}_{\pm 2.36}$ | $\mathbf{77.93}_{\pm 1.43}$ | $\mathbf{87.96}_{\pm 0.78}$ | $\mathbf{63.54}_{\pm 6.66}$ | $81.65_{\pm 2.62}$ | $\mathbf{52.05}_{\pm 0.75}$ |

