# OpenReview forum: "Risk Aware Negative Sampling in Link Prediction"
_ICLR.cc/2025/Conference — Submitted to ICLR 2025_

### Official Review · Reviewer_AazQ · 2024-10-22

**Soundness:** 2
**Presentation:** 3
**Contribution:** 2
**Rating:** 3
**Confidence:** 4

**Summary:**

This paper studies the negative sampling in link prediction. Different from previous studies focusing on developing more expressive models for link prediction, this study dives into the selection of negative samples and how a proper selection can maximize the potential performance of any parameterized link prediction models. It introduces Risk Aware Negative Sampling (RANS), which can effectively improve various GNN-based link prediction models by iteratively sampling harder negative node pairs from the graph.

**Strengths:**

[S1] The paper has a clear motivation for studying the effect of negative sampling on link predictions.

[S2] The proposed negative sampling method is algorithmically simple and effective.

[S3] The experimental results show that RANS can improve the performance of various link prediction models.

**Weaknesses:**

[W1] The scalability is a big issue, which makes RANS almost inapplicable to any large-scale graphs. If I understand correctly, the step 2 in the Algorithm would cost at least $O(N^2)$ for each training epoch. This is not acceptable for any link prediction models if they have the ambition to be applied in the real world.

[W2] Following W1, this makes RANS hard to evaluate on OGB link prediction datasets, including Collab, PPA, and Citation2. These benchmarks have become the commonly accepted testbeds for link prediction methods.

[W3] The connection between theoretical analysis and the methodology is weak. In fact, the theoretical analysis section (Section 5) has a minor contribution. Its analysis has nothing related to the task at hand, which is link prediction. The conclusion in the analysis is too general to focus on its impact on the link prediction task.

**Questions:**

* Even though the study of negative sampling for link prediction is both novel and interesting, the implementation part is too inefficient to be applied to many real-world use cases. Do authors consider any other efficient way to select the harder samples?

---

> ### Author Response · Authors · 2024-11-24
>
> We thank you for your detailed review of our work. Your suggestions have helped us identify areas where we can improve both the clarity and depth of our work, and we are committed to addressing these points in our revision.
>
> **W1:** We thank the reviewer for pointing this out. We believe that there may have been a misunderstanding about how RANS works algorithmically, and have updated the paper to make this point more clear. In particular, during each RANS resampling procedure, we draw a fixed number of negatives. If we ultimately desire $N$ negatives, we draw $kN$ negatives where $k>1$. These negatives can be drawn from any distribution, and in our case that distribution is the uniform distribution. In practice, $kN << N^2$. RANS does require $kN$ model calls, but graph models tend to be quite small. The linear number of model calls is similarly shared by MCNS, one our of baselines, that has enjoyed industrial usage.
>
> In addition to updating the prose of the paper to clarify this point, we have added a table comparing runtimes in the appendix. We observe that RANS is competitive from the perspective of computational cost with the other baselines presented, while providing superior performance.
>
> **W2:** We agree that while the presented results are promising, the datasets are small. We have added results for both ogbl-ddi and ogbl-collab, and have included them in Table 1. In both we observe significant performance improvements.
>
> **W3:** We believe that the paper is link prediction focused in at least two ways. The first is that, in the theoretical analysis, we consider the case where our number of negatives drastically outweights our positives (eg $N^2$ vs $N$), as is common in link prediction. Furthermore, our theoretical analysis that follows in Section 5 turns on some of the implicit assumptions of link prediction. Namely, there exist "easy" and "hard" samples, with "easy" samples far outweighing the hard ones. Beyond this, we note that the intuition the construction of RANS is based in link prediction as well. $\eta$ controls how many negatives need to be easy before we re-sample. An $\eta$ near one corresponded to a situation where most of the information had been "extracted" from the negatives. The intuition for $\delta$ is more link prediction inspired. Essentially, once the positive and negative edges have been separated, there is limited predictive benefit to "run up the score" by becoming even more confident about edges you already accurately categorize.
>
> It is true that some industrially relevant tasks such as recommendation systems have similar properties, but these comprise a graph as well, albeit implicitly, constructed from users, items, and their interactions [1,2].
>
> We feel as if we have addressed your questions and the weaknesses that you have pointed out. If we have not addressed them to your satisfaction, please do let us know. Otherwise, we would kindly request that you consider raising your score.
>
> [1] Aditya Pal et al. PinnerSage: Multi-Modal User Embedding Framework for Recommendations at Pinterest
>
> [2] Ahmed El-Kishky et al. TwHIN: Embedding the Twitter Heterogeneous Information Network for Personalized Recommendation

---

> > ### Comment · Reviewer_AazQ · 2024-11-25
> >
> > Thanks for the authors' responses to my questions. My concern about the scalability is resolved. However, I still wonder, regarding the OGB benchmark, why not include PPA and Citation2? Collab and ddi are two smaller ones compared to the other two. It will be much more convincing if the authors can include large-scale datasets.

---

> > > ### Author Response · Authors · 2024-11-26
> > >
> > > Rather than add experiments for PPA and Citation2, we opted to focus on adding experiments for an additional GNN backbone. We have completed those experiments, and have presented them in Table 6 in the appendix. On 5/6 datasets we find that RANS provides lifts above other baselines. We hope that this addresses your concerns about the lack of evidence to support RANS as a general training augmentation. With the inclusion of Table 6 in addition to the other additions to the work, we feel as if we have addressed your questions and the weaknesses that you have pointed out. If we have not addressed them to your satisfaction, please do let us know.

---

> > > > ### Comment · Reviewer_AazQ · 2024-11-27
> > > >
> > > > I appreciate the authors' responses to my concerns. However, the lack of experiments on OGB datasets, especially large-scale graphs like PPA and Citation2, hurt the empirical evidence of the work. Therefore, I think the paper needs another round of review. I encourage the authors to include results on those datasets in the revised version.

---

> > > > > ### Author Response · Authors · 2024-11-27
> > > > >
> > > > > Thank you for your feedback. We just wanted to clarify that we have included experiments for both ogbl-ddi and ogbl-collab in Table 1.

---

### Official Review · Reviewer_GxST · 2024-10-30

**Soundness:** 3
**Presentation:** 1
**Contribution:** 2
**Rating:** 5
**Confidence:** 5

**Summary:**

This paper proposes the Risk-Aware Negative Sampling (RANS) method for GNN-based link prediction. RANS dynamically samples hard negatives based on model predictions, with the goal of reducing excess risk.

**Strengths:**

- This paper unveils the significance of negative sampling in link prediction tasks, providing both empirical and theoretical evidence to support its importance.
- Experimental results demonstrate that RANS achieves notable performance improvements.

**Weaknesses:**

**W1.** Although the authors provide a theoretical analysis to highlight the importance of negative sampling, the methodological novelty may be insufficient for ICLR, as the proposed approach merely involves sampling hard negatives based on model predictions. Moreover, the method lacks a specific focus on link prediction and could be readily applied to other domains and tasks, which is not a tailored solution.

**W2.** The paper requires more thorough proofreading, as several critical issues diminish the overall quality.

Examples include:
   1. Algorithm 1:
      - The algorithm seems to be missing details, particularly around Line 6.
      - In Line 4, shouldn’t it be $N_{\text{easy}} = \sum_{\text{neg}} \mathbb{I}(p < \delta)$ instead of $\mathbb{I}(p > \delta)$? The easy negatives would be those with low scores from the link predictor.
      - The combination of mathematical notation and pseudo-code elements feels cluttered.
      - The inclusion of the `argsort` operation appears redundant, as the sorted indices do not seem to be used for filtering or refining ${\tilde{D}^n}^\prime$.
      - It’s unclear how many samples are selected as new negatives, making it difficult to infer from the algorithm table. Additionally, a more detailed description of the proposed methodology in Section 6 would be beneficial.

   2. What does $\tilde{D}_{p,n}$ in line 318 indicate?

   3. The caption for Figure 1 should be revised; the central ring atoms are actually represented as *larger* vertices.

**W3.** The paper's focus is exclusively on GCNs in both theoretical and empirical analyses, which limits the scope of applicability. The authors should provide experimental results with other backbone GNNs to verify RANS's versatility.

**W4.** The authors do not provide an analysis of hyperparameter sensitivity. Although RANS appears computationally efficient compared to subgraph GNNs, its practical applicability could be compromised if performance varies significantly with changes in parameters like $\eta$ and $q$.

**W5.** The paper omits important and popular link prediction benchmarks, such as OGBL-COLLAB and OGBL-DDI [1]. Experiments are only conducted on small-scale datasets.

---
**Reference**

[1] Open graph benchmark: Datasets for machine learning on graphs, NeurIPS 2020

**Questions:**

See weaknesses.

---

> ### Author Response · Authors · 2024-11-24
>
> We thank you for your detailed review of our work. Your suggestions have helped us identify areas where we can improve both the clarity and depth of our work, and we are committed to addressing these points in our revision.
>
> **W1:** We believe that the paper is link prediction focused in at least two ways. The first is that, in the theoretical analysis, we consider the case where our number of negatives drastically outweights our positives (eg $N^2$ vs $N$), as is common in link prediction. Furthermore, our theoretical analysis that follows in Section 5 turns on some of the implicit assumptions of link prediction. Namely, there exist "easy" and "hard" samples, with "easy'' samples far outweighing the hard ones. Beyond this, we note that the intuition the construction of RANS is based in link prediction as well. $\eta$ controls how many negatives need to be easy before we re-sample. An $\eta$ near one corresponded to a situation where most of the information had been "extracted" from the negatives. The intuition for $\delta$ is more link prediction inspired. Essentially, once the positive and negative edges have been separated, there is limited predictive benefit to "run up the score" by becoming even more confident about edges you already accurately categorize.
>
> **W2:** We thank the reviewer for pointing out these mistakes, and have fixed them in addition to performing a thorough proof reading of the paper. We apologize for these oversights.
>
> **W3:** There are no limitations in applying RANS to other methods. We opted to experiment with GCN due to its simplicity, the industrial relevance, and the desire to not overly complicate Table 1 by considering the Cartesian product of multiple GNN backbones and multiple sampling strategies. Additionally, we believe that our theoretical analysis is more general than just applying to GCNs. While Section 4 makes heuristic arguments with a GCN, this is to make our intuitive arguments for the importance of negative samples clear. Section 5 assumes, simply, that one has a class of learnable predictors. With that being said, we are running experiments HL-GNN+RANS [2] and will update the revision with those results if time permits.
>
> **W4:** Thank you for pointing this out. While both, $\eta$ and $\delta$ are fixed at runtime and we have performed no formal hyperparameter tuning over them. However, we believe that better understanding their influence on our algorithm is quite important. To this end, we have performed a scan of $\eta$ and $\delta$ and plotted the resulting surface in the appendix. We observe a broad region of good performance, indicating that our method does not require extensive hyperparameter tuning for use.
>
> **W5:** We agree that while the presented results are promising, the datasets are small. We have added results for both ogbl-ddi and ogbl-collab, and have included them in Table 1. For both datasets we observe meaningful lifts.
>
> We feel as if we have addressed your questions and the weaknesses that you have pointed out. If we have not addressed them to your satisfaction, please do let us know. Otherwise, we would kindly request that you consider raising your score.

---

> > ### Comment · Reviewer_GxST · 2024-11-25
> > **Official Comment by reviewer GxST**
> >
> > Which dataset was used to plot Figure 3? It is difficult to evaluate the robustness of RANS without knowing the performance of the best baseline. Moreover, the authors’ claim that RANS exhibits high robustness is hard to support based on the figure, as the non-red regions occupy nearly half of the landscape. Could the authors provide a 2D heatmap plot showing the difference in performance relative to the baseline? This would make it easier for clear assessment.

---

> > > ### Author Response · Authors · 2024-11-27
> > >
> > > We used Cora to construct Figure 3. We will generate the figure that you requested tomorrow and upload a new revision of the manuscript then.
> > >
> > > Additionally, we would like to draw your attention to new experiments that have been included in the paper. In Table 1 we have added results for both ogbl-ddi and ogbl-collab. We have also added Table 6 in the appendix that reports our experiments with HLGNN+RANS, and on 5/6 datasets we find that RANS provides lifts above other baselines. HLGNN was published at KDD this year and performed extremely well in link prediction settings.

---

### Official Review · Reviewer_VGoR · 2024-11-04

**Soundness:** 2
**Presentation:** 3
**Contribution:** 2
**Rating:** 5
**Confidence:** 4

**Summary:**

While recent advancements like labeling tricks, structural features, and subgraph techniques have improved link prediction performance, this paper investigates the potential benefits of using hard negative samples to further enhance model accuracy. The authors present a theoretical analysis from an empirical risk perspective, and introduce Risk Aware Negative Sampling (RANS). This method can be integrated with any MPNN model to improve training effectiveness. Experimental results demonstrate that a GCN model equipped with RANS achieves 20% to 50% higher predictive accuracy compared to models trained with standard negative sampling.

**Strengths:**

1. It provides a theoretical analysis from the empirical risk perspective to demonstrate that using hard negative samples could enhance the performance.

2. The experiment shows that a simple GCN plugged with the proposed method could advance the model performance, and behave better than other hard negative sampling methods.

**Weaknesses:**

1.While the analysis is interesting, the idea that hard negative sampling could improve model performance is somewhat expected. Additionally, the paper lacks a clear comparison between the proposed method and existing hard negative sampling approaches. Further analysis and discussion could help clarify the advantages of RANS over existing methods.

2.Some claims in the paper require additional clarification. For example, the statements made in lines 213-239 would be more convincing with empirical evidence or supporting citations.

3.The proposed method appears to have high computational complexity, which could limit its scalability on large datasets.

4.Including results on Open Graph Benchmark (OGB) datasets, such as ogbl-collab, ogbl-ppa, and ogbl-citation2, would strengthen the evaluation and make the results more generalizable.

5.While the experiments demonstrate that RANS improves GCN performance, it would be valuable to see if it can also enhance other popular methods, such as SEAL, BUDDY, Neo-GNN, ELPH, etc.

**Questions:**

1.What's the advantage of the RANS comparing with the existing hard negative sampling methods.

2.Do you have empirical evidence or supporting citations to support claims in lines 213-239?

3.What's the computational complexity of the RANS? Does it has the scalability issue for large datasets?

4.What are the results on OGB datasets?

5.What are the results on other popular methods using RANS, such as SEAL, BUDDY, Neo-GNN, ELPH?

---

> ### Author Response · Authors · 2024-11-24
>
> We thank you for your detailed review of our work. Your suggestions have helped us identify areas where we can improve both the clarity and depth of our work, and we are committed to addressing these points in our revision.
>
> **W1/Q1:** Many existing hard negative sampling methods are GAN based, which can be difficult to stabilize during training. The only hard negative sampling strategy that we know of that enjoys industrial adoption is MCNS, which we have included as a baseline. Compared to MCNS, we observe performance improvements across the board without significant model complexity, and those results are found in Table 1. Additionally, MCNS has been shown to outperform GAN based hard negative mining systems [1], and therefore we believe that it is a fair baseline to compare against for hard negative sampling.
>
> **W2/Q2:** The text from 213-239 was intended to be an illustrative example to provide some intuition as to why negative sampling might matter. With this in mind, we have added citations where appropriate. For example, we have cited the structurally isomorphic node problem. For the claim that negative sampling does not change the expressivity of the hypothesis space, we have added citations to indicate that the expressivity of the hypothesis space is data independent. In this context, expressivity specifically refers to what a model structure can represent. Our negative sampling procedure does not change that -- it simply helps improve the final model that is learned.
>
> **W3/Q3:** We thank the reviewer for pointing this out. We believe that there may have been a misunderstanding about how RANS works algorithmically, and have updated the paper to make this point more clear. In particular, during each RANS resampling procedure, we draw a fixed number of negatives. If we ultimately desire $N$ negatives, we draw $kN$ negatives where $k>1$. These negatives can be drawn from any distribution, and in our case that distribution is the uniform distribution. In practice, $kN << N^2$. RANS does require $kN$ model calls, but graph models tend to be quite small. The linear number of model calls is similarly shared by MCNS, one our of baselines, that has enjoyed industrial usage. We have updated the algorithmic discussion to make this point more clear. We have additionally included Table 5 in the appendix, which captures the total training time for all methods across all presented datasets. We observe that RANS is competitive with other dynamic sampling techniques in terms of runtime cost.
>
> **W4/Q4:** We have added results for both ogbl-ddi and ogbl-collab, and have included them in Table 1. For both datasets we observe meaningful lifts.
>
> **W5/Q5:** There are no limitations in applying RANS to other methods. We opted to experiment with GCN due to its simplicity, the industrial relevance, and the desire to not overly complicate Table 1 by considering the Cartesian product of multiple GNN backbones and multiple sampling strategies. With that being said, we are running experiments HL-GNN+RANS [2] and will update the revision with those results if time permits.
>
> We feel as if we have addressed your questions and the weaknesses that you have pointed out. If we have not addressed them to your satisfaction, please do let us know. Otherwise, we would kindly request that you consider raising your score.
>
> [1] Zhen Yang et al, Understanding Negative Sampling in Graph Representation Learning
>
> [2] Juzheng Zhang et al, Heuristic Learning with Graph Neural Networks: A Unified Framework for Link Prediction

---

> ### Author Response · Authors · 2024-11-26
>
> **W5/Q5:** We have added Table 6 in the appendix that reports our experiments with HLGNN+RANS, and on 5/6 datasets we find that RANS provides lifts above other baselines. We hope that this addresses your concerns about the lack of evidence to support RANS as a general training augmentation. With the inclusion of Table 6 in addition to the other additions to the work, we feel as if we have addressed your questions and the weaknesses that you have pointed out.
>
> We feel as if we have addressed your questions and the weaknesses that you have pointed out. If we have not addressed them to your satisfaction, please do let us know. Otherwise, we would kindly request that you consider raising your score.

---

> > ### Comment · Reviewer_VGoR · 2024-11-27
> > **response to the author**
> >
> > Thanks for the response! I still have the following concerns:
> >
> > 1. It seems that some related works, such as [1], might not have been referenced. Could you elaborate what's the advantage of  the proposed method compares to this work?
> >
> > [1] Wang et al. Not All Negatives Are Worth Attending to: Meta-Bootstrapping Negative Sampling Framework for Link Prediction, WSDM 2024.
> >
> > 2. I am still a bit unclear on how the proposed method relates to the concept of risk. It seems that RANS determines easy samples using $p$ in Algorithm 1. Could you clarify whether $p$ represents the probability or score of an edge? Additionally, the code could be made clearer; for example, what exactly does $D_{hardest}$ refer to?
> >
> > 3. I believe it would be valuable to demonstrate RANS applied to more advanced models, as GCN is a relatively simple backbone. Showing that RANS can help more sophisticated models achieve better performance would make the results even more compelling and convincing.
> >
> > 4. Regarding Table 6, I noticed that the collaboration result is approximately 52, while the HLGNN paper reports a value around 68. Could you clarify why there is such a significant difference?

---

> ### Author Response · Authors · 2024-11-27
>
> 1. Thank you for bringing this to our attention. We will include it in our literature review, and will describe any differences/advantages tomorrow.
>
> 2. $D^{n'}{hardest}$ denotes the set of negatives in $D^{n'}$ that are most challenging for the current model, specifically those negative samples that receive the highest model scores (indicating where the model is most incorrect). Conversely, $D^{n}{easiest}$ represents the negative samples that the model classifies most accurately. The model's score is denoted by $p$.
>
> Regarding RANS's relationship to risk: Our method was developed to minimize $\tilde{R}$, which theoretically requires evaluation on the complete dataset $D$. However, this is computationally infeasible for all but the smallest graphs. Therefore, we must sample $D$ to obtain $\hat{D}$ and consequently $\hat{R}$. This sampling introduces a fundamental challenge: minimizing $\hat{R}$ does not guarantee minimization of $\tilde{R}$. Consider a scenario where $\hat{D}^n$, where the superscript indicates the negative subset of the dataset, consists solely of trivial negatives—the risk gap could be substantial. The optimal scenario would be the opposite: if we constructed $\hat{D}^{opt, n}$ exclusively from the hardest negatives for the Bayes classifier $f^*$, then finding $\hat{f}$ that minimizes $\hat{R}$ would yield the best possible estimate for $f^*$ (Theorem 5.2).  While this assumes access to $f^*$, which is generally unavailable in practice, we address this limitation by using $f^{(n)}$ (where $n$ is the SGD index) as our current best estimate of $f^*$. This allows us to systematically replace "easy" negatives with "hard" ones, approximating $\hat{D}^{opt, n}$. By minimizing $\hat{R}^{opt}$ over $\hat{D}^{opt, n}$, we provide the most robust estimate possible for $f^*$.
>
>
> 3. Thank you for pointing out this concern. We would direct your attention to Table 6 in the appendix of the revised paper which reports the results for HLGNN. HLGNN was published at KDD this year and performed extremely well. We believe that HLGNN is a much more complicated backbone than a GCN, and hope that it addresses this point.
>
> 4. The performance differences can be attributed to several key factors. First, our experimental protocol differs in that we excluded validation edges from the testing phase, whereas HLGNN incorporated them. HLGNN's architecture learns fine-grained structural heuristics and exhibits sensitivity to edge count variations. Additionally, HLGNN features numerous hyperparameters beyond the core model architecture, including loss functions[1], learning rate schedules, and optimizer configurations. Given the time constraints of the rebuttal period, we maintained the default model hyperparameters from the provided codebase, implementing it within our experimental framework derived from the OGBL codebase, using standard BCE loss, Adam optimizer, and a fixed learning rate schedule.
>
> [1] AUC, HingeAUC, WeightedAUC, AdaptiveAUC, AdaptiveHingeAUC, logloss, CrossEntropyLoss, InfoNCE

---

> > ### Comment · Reviewer_VGoR · 2024-11-27
> > **response to the author**
> >
> > Thanks for the response!
> >
> > 1. If I understand correctly, it seems that RANS first randomly samples some edges and then selects the hardest ones based on the model's prediction scores. Is this interpretation accurate? If so, does the method still fundamentally rely on random sampling? Could you elaborate on whether this type of random sampling is sufficient to generate good candidate negative samples? I understand that generating scores for all possible negative edges would be infeasible due to the computational expense, but some clarification here would be helpful.
> >
> > 2. Regarding HLGNN, it might be beneficial to reproduce the results and provide a comparison with and without RANS. I understand that the response period is limited, but including such comparisons would make the evaluation more convincing. Additionally, demonstrating that RANS can benefit other advanced models, such as Buddy, Neo-GNN, or NCN, would further highlight its advantages and strengthen the impact of the work.

---

> > > ### Author Response · Authors · 2024-11-27
> > >
> > > You are correct that RANS requires generating new negatives according to some base distribution. We generate more negatives than we need, and score those. If we need N negatives, then we generate kN and score them all. From there, we select only the N hardest negatives and use those for training. We can use any base distribution we would like to generate these negatives, but we make use of the uniform distribution. The method depends on forward calls of the model being relatively cheap, but this is commonly the case for graph models.
> > >
> > > Thank you for your feedback on the inclusion of other models such as Buddy, Neo-GNN, or NCN. We unfortunately will not be able to complete these experiments before the deadline for revisions.

---

> > > > ### Comment · Reviewer_VGoR · 2024-11-27
> > > > **response to the author**
> > > >
> > > > Thank you for your response!
> > > >
> > > > From my understanding, it seems the approach selects hard negatives from a set of randomly sampled nodes. However, random sampling may include nodes that are completely irrelevant to each other. Given that the optimization goal of link prediction is to assign high scores to locally connected nodes and low scores to irrelevant ones, I am curious about the distinguishability of easy versus hard negatives under this strategy. Specifically, since random sampling often results in highly irrelevant nodes, especially in large graphs, how effective is this approach in identifying meaningful hard negatives? It would strengthen the work if the authors could provide empirical evidence to support this choice.

---

> > > > > ### Author Response · Authors · 2024-11-27
> > > > >
> > > > > You're correct that some of the negatives will indeed be easy, this is why we oversample. The logic is that if we generate K-times more negatives than we ultimately need, then we can score these and let the model scores decide which ones are truly easy. In this way, the approach is similar in spirit to a rejection sampling technique.
> > > > >
> > > > > As for the efficacy of this technique, we would direct your attention to Figure 2. In this figure, we observe that after resampling our train error drops precipitously.

---

> > > > > > ### Comment · Reviewer_VGoR · 2024-11-27
> > > > > > **response to the author**
> > > > > >
> > > > > > Thank you for the response! I understand that when the candidate set is large, meaningful hard negative samples can be selected. However, it inevitably introduces scalability challenges, particularly for large graphs. This creates a tradeoff between the effectiveness of hard negative samples and training efficiency. It would be valuable to include an analysis of this tradeoff to provide further insights.
> > > > > >
> > > > > > Additionally, could you clarify the number of negative samples used for each dataset? Do the baselines also use the same number of negative samples? I noticed that the reported results for BUDDY are consistent with its original paper, and the source code for BUDDY appears to use only one negative sample. If this is the case, it might lead to an unfair comparison with the baselines.

---

> > > > > > > ### Author Response · Authors · 2024-11-27
> > > > > > >
> > > > > > > For all Planetoid and WikipediaNetwork datasets we used an oversampling factor of $k=10$. For the two ogbl datasets we used an oversampling factor of $k=2$. We have implemented a naive implementation that generates all candidate negatives first, and then scores them all. An optimal solution would do this in batches using a priority queue. Due to the time constraints of the rebuttal period, we prioritized finishing the requested experiments over implementing a more compute and memory optimal solution. Despite oversampling, we select only enough negatives such that we only train on N negatives, where N is the number of positive edges, such that the dataset is perfectly balanced.

---

> > > > > > > > ### Comment · Reviewer_VGoR · 2024-11-27
> > > > > > > > **response to the author**
> > > > > > > >
> > > > > > > > Thank you for the clarification and for engaging in the discussion! I would recommend revising the paper for submission to another venue with the following enhancements to strengthen its contributions and impact:
> > > > > > > >
> > > > > > > > 1. Include more analysis on the tradeoff between training efficiency and the effectiveness of hard negative sampling.
> > > > > > > >
> > > > > > > > 2. Provide results on larger datasets.
> > > > > > > >
> > > > > > > > 3. Results of more advanced models.
> > > > > > > >
> > > > > > > > 4. Additionally, I suggest including a broader review of related works beyond the link prediction domain. Hard negative sampling is a well-studied topic in recommendation systems, which is also a link prediction task. Reviewing these methods and thinking of the novelty or advantages of the proposed approach would add more value to the paper.

---

### Official Review · Reviewer_Aaxd · 2024-11-04

**Soundness:** 2
**Presentation:** 3
**Contribution:** 3
**Rating:** 6
**Confidence:** 4

**Summary:**

The authors propose a new methodology for negative sampling that considers theoretically provable assumptions about excess risk in the supervised link-prediction (LP) setting to enhance expressiveness and therefore performance of LP models. From which, the theorems and proofs are integrated into a clean algorithm which is applied in experiments that demonstrate: enhanced performance versus other sampling methods, more practical alignment between training/validation/testing performance, demonstrable closing of the performance gap between negative sampling strategies, and competitive performance for simple baselines versus SOTA LP methods.

**Strengths:**

* The authors take great care to relate their work to existing methodologies, such as expressive sampling methods and even other architectures like GANs. This flows nicely into the theoretical assumptions and justifications, providing a solid foundation from which the proposed RANS relates to other works.
* The proofs used for the intuition behind constructing the RANS algorithm are exceptionally clear, the added context is helpful for quantifying the impact of the theoretical assumptions and the logical flow is intuitive as to what sort of contribution that RANS provides.
* The authors make use of experimental methodologies that are also intuitive and relate well to existing research on sampling strategies, coupled with the use of the HR@K metric adds robustness to the performance gains provided by RANS in all experiments conducted for this paper.

**Weaknesses:**

* The current use of smaller benchmark datasets is of great practical concern, especially given that the risk associated with sampled negative edges is dependent on an estimation. Do we see a similar level of performance improvement when testing with larger benchmarking datasets, such as the OGB datasets [1]? Without such a test, it is difficult to determine whether replacing easy negative samples with the risk-aware negative samples at a given epoch is not only a safe assumption but whether or not the theoretical assumptions hold in larger sample spaces.
* I understand that maintaining fixed hyperparameters and model is important for testing just the sampling strategies, but given the relatively high-level of variance within the results for each table, especially Table 1, it seems pertinent to run a significance test between the scoring distributions to provide statistical confidence on potential performance gains across sampling strategies.
* The results in Table 3 are difficult to believe given that GCN+RANS is the untuned variant applied in previous experiments. Have the SOTA LP methods been tuned on the cora and citeseer datasets? This becomes especially critical given that lack of tests on larger benchmark datasets coupled with RANS theoretical assumption.
* What sort of restrictions might there be on applying RANS to SOTA LP methods such as BUDDY or SEAL? Given the performance enhancement RANS provides to GCN, testing potential gains that RANS can provide to the more-expressive SOTA LP architectures could provide strong empirical evidence on RANS effectiveness, eliminating concerns about it's practical applicability and reliance on risk-estimation.

[1] Hu, Weihua, Matthias Fey, Marinka Zitnik, Yuxiao Dong, Hongyu Ren, Bowen Liu, Michele Catasta, and Jure Leskovec. "Open graph benchmark: Datasets for machine learning on graphs." Advances in neural information processing systems 33 (2020): 22118-22133.

**Questions:**

* See "Weaknesses" Section of the review for the most-pertinent questions.
* Are the nu and delta hyperparameters in Algorithm fixed at runtime? If so, it is not possible to tune the hyperparameters to better understand their effect on performance or even expressivity?
* What is the intuition for setting nu to 0.95 and delta to 0.1? Is this just to restrict the sampled search space without being too restrictive? Considerations about these assumptions could be used for additional experiments on better understanding negative sampling, similar to what was detailed in Table 2.
* The following is not critical to the current review. Given the increased research interest in negative-sampling for link-prediction, article [2] may be of interest to the authors, especially the experiments regarding the alignment of positive and negative sample distributions.
[2] Nguyen, Trung-Kien, and Yuan Fang. "Diffusion-based Negative Sampling on Graphs for Link Prediction." In Proceedings of the ACM on Web Conference 2024, pp. 948-958. 2024.

---

> ### Author Response · Authors · 2024-11-24
>
> We thank reviewer AAXD for their thorough and thoughtful review of our work. We feel that by addressing their points, our work has become stronger and clearer. Specifically:
>
> **W1:** We agree that while the presented results are promising, the datasets are small. We have added results for both ogbl-ddi and ogbl-collab, and have included them in Table 1. For both ogbl-ddi and ogbl-collab, we observe statistically significant lifts.
>
> **W2:** The results in Table 3 for GCN+UNS, SEAL, BUDDY, ELPH, and Neo-GNN are replicated [1], and as such they correspond to the optimal hyperparameters for the optimal models. For GCN+RANS, we have performed no such hyperparameter tuning. We have added citations and prose to make this more clear.
>
> **W3:** There are no limitations in applying RANS to other methods. We opted to experiment with GCN due to its simplicity, the industrial relevance, and the desire to not overly complicate Table 1 by considering the Cartesian product of multiple GNN backbones and multiple sampling strategies. With that being said, we are running experiments HL-GNN+RANS [2] and will update the revision with those results if time permits.
>
> **Q1:** Thank you for pointing this out. Yes, $\eta$ and $\delta$ are fixed at runtime and we have performed no formal hyperparameter tuning over them. However, we believe that better understanding their influence on our algorithm is quite important. To this end, we have performed a scan of $\eta$ and $\delta$ and plotted the resulting surface in the appendix. We observe that there is a wide region of stability for our method, which means only limited tuning is required.
>
> **Q2:** We selected $\eta$ to be 0.95 because we thought that this corresponded to a situation where most of the information had been "extracted" from the negatives. The intuition for $\delta$ is more link prediction inspired. Essentially, once the positive and negative edges have been separated, there is limited predictive benefit to "run up the score" by becoming even more confident about edges you already accurately categorize.
>
> **Q3:** We thank the reviewer for pointing out this article, as we had missed it and have added it to our literature review. We have updated our literature review to include this work.
>
> We feel as if we have addressed your questions and the weaknesses that you have pointed out. If we have not addressed them to your satisfaction, please do let us know. Otherwise, we would kindly request that you consider raising your score.
>
> [1] BP Chamberlin et al, GRAPH NEURAL NETWORKS FOR LINK PREDICTION WITH SUBGRAPH SKETCHING
>
> [2] Juzheng Zhang et al, Heuristic Learning with Graph Neural Networks: A Unified Framework for Link Prediction

---

> > ### Comment · Reviewer_Aaxd · 2024-11-25
> >
> > Thank you for your response, concerns about why the nu and delta parameters were fixed have been addressed. However there are concerns that remain:
> > 1. It is understandable that elevating more advanced models with RANS is complicated. However, the claim that RANS can be integrated into more advanced models needs evidence to indicate such an inclusion is possible. Conducting experiments with HL-GNN+RANS is sufficient to indicate this, even if its only on the smaller datasets.
> > 2. What is meant by 'left' and 'right' to indicate the regions for hyperparameter sensitivity of RANS and UNS in the Figure 3. It seems that the effect from RANS is indicated in red and the effect from UNS is indicated in blue? Does that mean UNS is also subjected to the nu and delta parameters? Why not make a subplot for each sampling method to clarify the distinction between their search spaces?
> > 3. The results in Table 1 are promising, but concerns remain about Table 6 results. Since GCN+RANS is not tuned specifically-tuned for Table 6 and the ogbl datasets are included in Table 1, then why not add relevant ogbl-collab (HR@50) and ogbl-ddi (HR@20) results for GCN+RANS to Table 6? Since GCN+RANS was used for industrial-relevance, then why not display the benefit that this methodology applies to larger benchmark datasets? The inclusion of those results would further indicate that simply including RANS to GCN could enhance it's performance closer to a standard set by the more advanced models on the larger datasets.
> > 4. Given what the paper is claiming, this is less of a concern and more of a question. It only seems natural to tune GCN+RANS to perform better within Table 6, was this considered? Or were the benefits provided by further tuning only marginal given the effect of RANS? Considerations for the second question seem important for applications of RANS. Additional tuning may be impractical due to current time-constraints, so it is not necessary for this review.
> >
> > Due to these concerns, I will keep my current score.

---

> > > ### Author Response · Authors · 2024-11-25
> > >
> > > **Q2:** Our apologies, it appears a stale version of Figure 3's caption was included with the pdf. We have updated the submission accordingly. We hope that resolves question 2. RANS is the only method subject to $\eta$ and $\delta$.
> > >
> > > **Q3:** We do not have a Table 6 -- could you clarify which table you are referring to? Is it Table 3? More to the point, we specifically wanted to avoid confounding model hyperparameters and negative sampling strategies which is why we opted not to tune for the webkb and planetoid datasets in Table 1. For OGBL-DDI and OGBL-COLLAB, we worked from the baseline code that was provided by the OGBL team [1], and used their default hyperparameters after confirming that their GCN+UNS implementation replicated the results presented on the leaderboard. Note, because the provided implementation performs uniform negative sampling during every minibatch, this is dynamic uniform sampling. In this respect, we are starting from near optimal model hyperparameters. As for including the OGBL datasets in Table 3, we believe that this is a great idea.
> > >
> > > **Q4:** The question is quite a natural one. To evaluate whether we would see significant performance improvements from a hyperparameter tune (both on RANS specific parameters, and GCN hyperparameters), we first performed an informal tune. Essentially, we varied $\eta$ and $\delta$ over a few runs with both CORA and CITESEER to select parameters that balanced the increased computational cost against the performance increases. We found empirically that RANS was robust to modest hyperparameter changes (a conclusion validated by Figure 3), so we opted not to pursue extensive RANS tuning. We did not opt to tune the GCN in Table 3 because we felt that it muddied the story. Doing such a tune makes it more challenging to potentially suss-out whether the performance increase comes from a better set of negative samples, a better set of hyper-parameters, or both. We valued the narrative simplicity and purity of experiment above the potential few percentage point improvements that we might have found from additional tuning.
> > >
> > > [1] https://github.com/snap-stanford/ogb/tree/master/examples/linkproppred/ddi

---

> > > > ### Comment · Reviewer_Aaxd · 2024-11-25
> > > >
> > > > Thank you to the authors for the clarification on: the hyperparameter tuning, Figure 3, and Table 3 (my typo said Table 6).
> > > >
> > > > I appreciate how crisp and clear the story is within the article. I understand that analysis is focused on GCN since it is much more efficient than many advanced LP models. However, BUDDY serves as a scalable alternative to many advanced LP models. Therefore, BUDDY's superior performance over GCN+RANS coupled with it's scalability reduces the industrial usefulness of GCN+RANS. Tuning GCN+RANS further, and then including it as another model, such as "Tuned GCN+RANS", into Table 3 would improve the claims in this paper significantly. This significance stems from a Tuned GCN+RANS providing evidence to support the claim about RANS's ability to close the generalization gap; effectively tying the Table 3 results back to Figure 2.
> > > >
> > > > My previous concern about a lack of evidence to support the claim that RANS serves as a training augmentation to any LP model remains. Is there evidence available to support RANS's flexibility?

---

> > > > > ### Author Response · Authors · 2024-11-25
> > > > >
> > > > > The experiments for HLGNN are running as we speak for the Planetoid, WikipediaNetwork, and OGBL datasets. The results are being compiled in a table that I will place in the appendix and reference from the main text. In addition, we will include the relevant HLGNN + UNS and HLGNN + RANS results in Table 3.

---

> ### Author Response · Authors · 2024-11-26
>
> First, we would like to thank the reviewer for being unusually active during the discussion period. We feel that the reviewers suggestions have resulted in a stronger paper.
>
> **W3:** We have added Table 6 in the appendix that reports our experiments with HLGNN+RANS, and on 5/6 datasets we find that RANS provides lifts above other baselines. We hope that this addresses your concerns about the lack of evidence to support RANS as a general training augmentation. With the inclusion of Table 6 in addition to the other additions to the work, we feel as if we have addressed your questions and the weaknesses that you have pointed out. If we have not addressed them to your satisfaction, please do let us know.

---

> > ### Comment · Reviewer_Aaxd · 2024-11-27
> >
> > Thank you to the authors for engaging with my review, I appreciate the candor along with the revised paper. My concerns are addressed and I have raised the review score. On a side note: I believe Table 1 and Table 6 should be called "Principle" and not "Principal"

---

### Meta-Review · Area_Chair_Fyoi · 2024-12-18

**Metareview:**

### Summary
The paper proposes Risk-Aware Negative Sampling (RANS), a method for improving link prediction (LP) performance in GNN models. RANS dynamically samples hard negatives during training based on model predictions, with a theoretical analysis from the empirical risk perspective to support its motivation. Experimental results on small-scale datasets show that RANS significantly improves LP performance, particularly when applied to GCNs.

### Strengths
- The paper highlights the importance of negative sampling in LP and demonstrates its impact through both theoretical analysis and experimental results.
- The proposed RANS algorithm is straightforward, practically enhancing LP performance with better negative sample selection.
- Experiments demonstrate notable performance gains for GCN-based models compared to standard negative sampling strategies.

### Weaknesses
- The computational complexity of RANS is high, with a potential O(N²) cost per epoch, making it infeasible for large-scale graphs.
The lack of evaluation on OGB benchmarks (e.g., Collab, PPA, Citation2) raises doubts about the method's applicability to real-world datasets.
- Experiments are conducted on small datasets, and the baselines include relatively outdated methods. The paper does not test RANS with modern SOTA LP models like SEAL, BUDDY, or Neo-GNN. Variance in the reported results (e.g., Table 1) raises concerns about the statistical significance of the improvements.
- The theoretical analysis is generic and not strongly tied to the specific task of link prediction. Its connection to the proposed methodology is weak. The impact of RANS parameters (e.g., $\eta$, $q$) is not explored, leaving uncertainty about its robustness and sensitivity to hyperparameters.

While the paper addresses a meaningful problem and proposes a promising method for improving link prediction through risk-aware negative sampling, it suffers from scalability issues, limited experimental evaluation, and a weak connection between theory and practice. Without testing on larger benchmarks and modern LP methods, the method's generalizability and practical applicability remain unclear. Further optimizations for scalability and comprehensive evaluations are necessary to strengthen the contributions.

**Additional Comments On Reviewer Discussion:**

The above weaknesses are the major concerns raised by the reviewers. However, the authors still did not perform experiments on large scale datasets like PPA and Citation 2, which limits the practicality of the proposed method.

---

### Decision · Program_Chairs · 2025-01-22

Reject